# Microplastic Formation and Surface Crack Patterns: A Method for Waste Plastic Identification

**DOI:** 10.3390/molecules30224461

**Published:** 2025-11-19

**Authors:** Hisayuki Nakatani, Anh Thi Ngoc Dao

**Affiliations:** 1Chemistry and Materials Engineering Program, Graduate School of Integrated Science and Technology, Nagasaki University, 1-14 Bunkyo-machi, Nagasaki 852-8521, Japan; anh.dao@nagasaki-u.ac.jp; 2Organization for Marine Science and Technology, Nagasaki University, 1-14 Bunkyo-machi, Nagasaki 852-8521, Japan

**Keywords:** microplastic formation, autoxidation, crack texture, polyolefin, degradation, seawater effect, AI-based sorting

## Abstract

Accumulation of plastic debris in marine environments has become a critical global issue, with microplastics (MPs) posing persistent ecological risks. This review synthesizes current knowledge on the formation mechanisms of MPs from polyolefins such as polypropylene (PP) and polyethylene (PE), emphasizing the influence of marine conditions on degradation pathways. Autoxidation is identified as the dominant mechanism; however, salinity and chloride ions significantly retard radical formation, altering photodegradation kinetics and crack propagation. These effects lead to distinctive surface morphologies—such as rectangular and trapezoidal crack patterns in PP—which can serve as reliable indicators for polymer identification. This review further explores the role of polymer chain orientation and spherulite structures in crack development and discusses how these features can be leveraged for cost-effective sorting and recycling strategies. Finally, emerging approaches using AI-based image recognition for automated identification of weathered plastics are highlighted as promising tools to enhance resource recovery and mitigate marine plastic pollution.

## 1. Introduction

The long-term use of several plastic materials has resulted in the accumulation of a large amount of litter in the marine environment [1,2,3,4,5,6,7,8,9,10,11,12,13,14,15]. Plastic debris has become one of the most significant environmental issues. Microplastics (MPs), which are formed from larger plastic debris, have raised growing environmental concerns [16,17,18,19,20]. Polyethylene (PE), polypropylene (PP), and polystyrene (PS) are widely produced worldwide due to their low densities. MPs float on the surface of the sea. Furthermore, they are not biodegradable, and thus remain in the marine environment indefinitely. The location of MP production has not yet been clarified. For example, some have been generated on land and others in the sea. In some cases, MPs partially leave the sea and enter the atmosphere [21]. They occur in various places and circulate. Several types of MPs have been reported to form in water through exposure to visible and/or UV light [22,23,24]. Crystalline polymers, such as PP and PE, consist of crystalline and amorphous parts. The complicated matrix certainly affects the rate of autoxidation-induced degradation. PP degradation is heterogeneous and complicated by the permeability of light, heat, oxygen, and the diffusibility of degradation initiators [25,26,27,28,29]. In amorphous polymers, such as polystyrene (PS), the degradation spreading behavior is uniform. Recrystallization, known as “chemical crystallization”, does not occur during degradation in crystalline polymers. The shape of MPs generated by degradation is expected to differ between crystalline and amorphous polymers.

The formation of MPs is certainly associated with autoxidation. Polymers containing carbon–carbon bonds degrade in the solid state due to autoxidation [22,23,24]. In addition, since the temperature at which MPs are generated under natural conditions is below 80 °C, the decomposition mode of the hydroperoxide group in autoxidation is single-molecule decomposition [30], resulting in a large amount of undecomposed hydroperoxide remaining and slow progress of autoxidation. The presence of vinyl and vinylidene groups in weathered polypropylene (PP) indicates that reactions other than autoxidation, such as the Norrish reaction, are occurring [31]. It is also known that differences in atmosphere, such as those in the air, underwater, and in seawater, affect the rate of autoxidation [32]. Since most marine MP is generated along coastlines [33,34], it forms under the influence of sunlight, seawater, and the mechanical effects of waves. In other words, it is exposed to a deteriorating environment that causes mechanical stress due to friction, triggers the Norrish reaction due to light irradiation, and is exposed to seawater, which suppresses autoxidation. The weathering degradation of PP and PE, which has been studied for many years, has been conducted on land, and the effects of seawater have not been considered at all. Therefore, in MP research, the degradation mechanism of these plastic products has been interpreted based on conventional autoxidation mechanism. It goes without saying that estimating the mechanism of MP generation due to degradation without considering the differences between land and marine environments will result in discrepancies from the actual generation mechanism. It should also be noted that degradation behavior is highly sensitive to temperature.

PP, PE and expanded polystyrene (EPS) are low-density plastics produced in large quantities worldwide. The debris floats on the sea surface or is washed ashore by waves, traveling between the sea and land [15]. Their surfaces are exposed to seawater, which is expected to affect degradation reactions. Seawater contains various mineral components and easily influences autoxidation, which is the main reaction in degradation reactions. The autoxidation behavior of these plastics has been studied in detail for many years. However, since the purpose of the research was to elucidate the weathering behavior of these plastic products, the possibility of their remaining as marine debris was not considered, and therefore their autoxidation behavior in seawater has not been investigate. The mechanism of MP formation is certainly related to degradation reactions such as autoxidation, but it is dangerous to blindly accept conventional research on degradation. Therefore, rather than relying solely on conventional weathering studies conducted in terrestrial environments, it is essential to reevaluate the degradation mechanisms of PP, PE, and EPS under actual marine conditions. The presence of salts, dissolved oxygen, and biological activity in seawater can significantly alter the autoxidation kinetics and pathways. This can lead to different degradation products and rates of MP formation. Additionally, the interaction between floating debris and marine organisms, as well as exposure to sunlight and wave action, introduces variables that must be considered. A comprehensive understanding of these factors is crucial for accurately predicting the environmental fate of plastics and developing effective strategies to mitigate marine plastic pollution.

This review introduces studies on crack patterns of various types of plastic marine debris surfaces caused by degradation and is based on a systematic literature search using predefined keywords in peer-reviewed articles from databases such as Scopus and Web of Science, covering studies on polyolefin degradation, crack morphology, and identification techniques for recycling. A total of approximately 80 studies published between 1960 and 2024 were analyzed, including both laboratory and field observations. Furthermore, Isobe et al. estimated that the average residence time of plastic debris in the upper ocean is approximately three years [15]. This value provides a useful benchmark for understanding the half-life of microplastic generation and removal in the marine environment, highlighting the importance of developing effective waste plastic sorting technologies—a necessary technique for plastic recycling discussed in this review.

## 2. Degradation Reactions and Progression Behavior

### 2.1. Autoxidation and Side Reactions

Autoxidation is a type of radical oxidation that occurs in hydrocarbon structures. These structures are common, and most polymers contain them to some extent. Therefore, it can be said that autoxidation is a degradation reaction that occurs in almost all polymers. In the case of polyolefins, such as polypropylene and polyethylene, which are composed solely of hydrocarbon structures, autoxidation is the only degradation reaction that occurs. As shown in Figure 1, autoxidation is initiated by the generation of alkyl radicals (R·) due to heat and light [25,26,35,36,37]. The generated R· reacts with oxygen to form peroxy radicals (ROO·), which extract hydrogen from the main chain to form hydroperoxides (ROOH). Subsequently, hydroperoxides decompose under the influence of heat or light. This leads to the generation of alkyl radicals (RO·) and the subsequent formation of ketones. These ketones cause the main chain to break. During chain scission, R· is regenerated, and the reaction cycle repeats, gradually reducing the molecular weight.

As degradation progresses, various side reactions occur, and the byproducts diversify [31]. For example, polypropylene (PP) exhibits a peak corresponding to ketone groups (methyl ketone) at around 1715 cm^−1^ in infrared absorption spectroscopy (IR) in the early stages of degradation. The peak intensity increases with degradation time, making it a useful indicator of degradation progression. Methyl ketone is produced by the autoxidation of the main chain, as shown in Figure 1. Therefore, it is ideal for determining the progress of autoxidation degradation. The increase in the peak intensity of the ketone group serves as an indicator of degradation progression. This intensity is normalized by dividing it by the peak intensity of the methylene group in the main chain. It is often used as an indicator called the “carbonyl index (CI)”. When oxidative degradation of polyolefin occurs, the autoxidation mechanism dominates the process. This allows the progression of degradation to be assessed using CI values.

However, as the degradation progresses, various side reactions occur, leading to the formation of diverse functional groups [31,38]. Esters, vinyl alkenes, acids, aldehydes, and γ-lactones, for instance, are produced. Table 1 summarizes the types of polyolefins in which these functional groups are produced using various degradation methods [31,39,40,41,42,43,44]. Fundamentally, the differences in degradation methods appear to have little effect on side reactions. Many functional groups originate from carbonyl compounds, but the side reactions they generate stem from hydroperoxide compounds formed by autoxidation reactions [45]. Different types of side reaction arise from the distinct reaction mechanisms that result from variations in how the hydroperoxide reaction proceeds. Since the side reactions originate from hydroperoxide, light or heat stimulation appears to only affect how easily each reaction occurs, at most influencing the ratio of carbonyl compound formation. In fact, the primary functional groups in the oxidation products of photodegraded PP are esters, followed by vinyl alkenes and acids [38]. In contrast, thermally oxidized polypropylene contains significantly fewer esters and vinyl alkenes, and relatively more aldehydes, ketones, and γ-lactones [38]. Of these, γ-lactone is particularly prone to formation via the back-biting process within the molecular chains of E-P copolymers [46]. Meanwhile, vinyl alkenes are easily generated through the Norrish reaction involving hydroperoxide or terminal carbonyl groups [47]. In any case, there is little difference in the types of byproducts generated by photodegradation and thermal degradation.

Although the formation of functional groups such as esters, acids, and vinyl alkenes during polyolefin degradation has been reported (see Table 1), the quantitative rate of their formation and the subsequent reactions—such as removal or utilization by organisms—remain largely unknown. Current studies provide qualitative evidence of side reactions but do not offer kinetic data or ecological assimilation pathways. Therefore, a detailed discussion on these aspects is beyond the scope of this review and represents an important area for future research. As shown in Figure 2, the infrared spectrum of degraded polypropylene reveals overlapping peaks corresponding to lactones, esters, ketones, and unsaturated double bonds. These side products, formed during advanced stages of degradation, complicate the accurate quantification of methyl ketone groups used to calculate the carbonyl index. Consequently, CI is a reliable indicator only in the early stages of degradation. In later stages, the accumulation of secondary oxidation products and overlapping spectral features significantly reduce its reliability. Moreover, in Py-GC-MS analysis, the presence of alkanes and alkenes derived from polyethylene and fatty substances can confound the identification of degradation markers, further limiting the utility of CI.

### 2.2. Non-Uniform Degradation Behavior of Polyolefins

The autoxidation of polyolefins and subsequent side reactions have been extensively studied [48]. However, the heterogeneous diffusion and propagation behavior of polyolefins in the solid state, particularly in amorphous and crystalline regions, remains unclear, and detailed research is ongoing. The degradation diffusion behavior of polyolefins during the photodegradation process was visualized by combining surface measurements using Raman mapping and SEM/EDX [49].

Currently, most polyolefin polymerization process is conducted via gas-phase polymerization without degassing. Consequently, catalyst residues, such as Ti compounds, are inevitably present in the polymer, although in trace amounts. Furthermore, trace amounts of hydroperoxide (ROOH) are generated within polyolefins during molding. Figure 3 shows the non-uniform degradation caused by metal catalyst residues and ROOH produced during molding. During photodegradation, the ROOH near metal particles decomposes via photocatalytic action. This decomposition generates radical species. When these species collide with the surrounding polymer matrix, they initiate autoxidation. The metal particles are transition metals characterized by the presence of low-valent species. These low-valent metal species reduce the radical species, converting them into non-radical species. Metal particles serve as ROOH decomposition sites, but they also function as radical scavengers. The high oxidation levels observed several micrometers away from the metal particles result from their specific effect, which contributes to the heterogeneous degradation behavior of polyolefins [49]. Studies on polyolefin degradation have generally been performed in molten and/or molten-like states [30,35,36,37]. In these states, degradation follows classical kinetics and is a homogeneous process. However, polyolefin materials are usually exposed to mild environmental conditions, such as sunlight and lower temperatures. In these cases, degradation occurs in the solid state [20,25,26,27,29,49], and the degradation process is heterogeneous. This heterogeneous degradation behavior is complicated by the permeability of light, heat, and oxygen; the diffusibility of degradation initiators; and the superstructure [20,26,27,29,49,50,51,52,53]. Furthermore, when considering polyolefin composites such as polyolefin/talc, the effects of filler on polyolefin degradation must be considered, as metallic compounds which are constituents and impurities of filler could accelerate degradation [54]. In other words, the need to consider localized degradation arises.

Cracks are expected to form near metal compound residues during the weathering process. However, the locations where cracks appear on typical material surfaces vary significantly. This is because the primary factors causing crack initiation are closely related to mechanical aspects, such as warping, strain, and molecular chain orientation. When degradation begins at a surface location where strain is applied, microcracks form perpendicular to the orientation direction [55]. These microcracks then coalesce and grow into macrocracks. In MP research, these macro-cracks are often referred to as linear fractures [56]. Other surface regions undergo degradation-induced changes as well. Detailed mechanical studies have shown that weathering-induced degradation leads to brittleness across the entire surface [57]. These studies have investigated: chemical crystallization, changes in the melting point, and microfracturing. The weathering degradation of polyolefins on land is a significant source of microplastics. Due to the abundance of prior research, the microfracturing mechanisms on land, particularly those forming microplastics, are not novel. However, the degradation behavior appears to differ when seawater is present, such as during weathering degradation on coasts, at sea, or under seawater [58,59,60].

### 2.3. The Effect of Seawater on the Degradation Behavior of Polyolefins

The potential toxicity of MPs may increase as particle size decreases. However, the particle size distribution, formation rates, and degradation rates are fundamentally unknown, making it extremely difficult to predict their environmental impact [7]. The primary location of MP production remains unclear. For example, one group was generated on land, and the others were generated in the sea. Although MPs are produced in various places, they are primarily considered to be produced in the sea and its surroundings. The mechanisms by which MPs are produced on land and at sea differ considerably [58]. Delamination was observed on the surface of MPs retrieved from the seashore. However, an abrasion patch structure, but no delamination marks, was observed on the MP surface retrieved from the riverside [58]. The generation of MPs is clearly associated with surface cracks resulting from degradation. The delaminated parts become much smaller MP particles and are released into the environment. The cracking pattern and texture are likely related to the ease of delamination. Table 2 summarizes the crack textures on various weathered or photo-degraded plastic surfaces under different environmental conditions [24,31,57,61,62,63,64,65,66]. For both PP and PE, the surface crack texture is strongly influenced by flow conditions during molding, such as flow direction and biaxial orientation (blow molding). The texture of PP differs from that of PE in that PP exhibits cracks running vertically and horizontally, forming a square-like pattern [61]. The presence of spherulite structures influences pattern formation [24]. In contrast, PE(LDPE) shows the formation of fragments, namely fine particles [65]. The presence of superstructure, such as globular crystals and gels [67,68,69,70], appears to significantly impact the texture of cracks that form during degradation.

Salinity lowers the degradation level of polyolefins, such as PP and PE [19,32,59,60]. Previous have shown indicate that fragmentation occurs not only in PP but also in PS and LDPE. However, the fragmentation rates vary significantly, and additional fragmentation mechanisms may operate in aquatic environments [58]. Variations in polymer type influence degradation pathways, superstructures, and resistance to autoxidation, all of which are believed to affect both the rate and mechanism of fragmentation. PP, PE, and expanded PS have low densities and therefore tend to float on the sea surface or wash ashore, where they are exposed to seawater and are degraded by sunlight. Therefore, the impact of seawater—that is, the presence of salt—on degradation must be carefully considered. Aqueous Cl^−^ functions as an inhibitor in autoxidation of polyolefin in seawater [32].

Figure 4 illustrates the mechanisms for transfer reaction to Cl^−^ in seawater and the formation of chlorine-containing species in Cl_2_ solution as a function of pH. In the photodegradation of polyolefins in seawater with a pH of approximately 8, Cl^−^ reacts with OH· generated by sunlight to form ClOH·^−^, and the equilibrium shifts toward the less reactive ClO^−^. This inhibitory effect of Cl^−^ suppresses the generation of OH· radicals, which act as initiators for oxidative decomposition (autoxidation) reactions. Consequently seawater acts as a retarder of autoxidation, thereby reducing the rate of photodegradation or weathering degradation. Previous studies have also investigated the degradation of polyethylene microplastics in seawater, highlighting the role of salinity and environmental conditions in slowing oxidative degradation pathways [71].

Table 3 summarizes the crack textures observed on various weathered or photo-degraded plastic surfaces under seawater conditions [19,56,72,73]. Compared to crack textures formed in the absence of seawater (see Table 2), distinct textures, such as conchoidal fractures, have been observed. Overall, however, they exhibit nearly identical textures. Similarly to what was observed without seawater, the surface texture reflects the molecular orientation and superstructures present during molding. The presence of seawater seems to affect the preservation of surface texture more than it affects the texture of the cracks themselves. The suppression of the autoxidation reaction by seawater slows the development of cracks and subsequent surface degradation. For PP, this suppression effect significantly contributes to maintaining surface texture. Cracks running parallel to the flow (injection) direction and those running perpendicular to them form the characteristic rectangular or trapezoidal patterns frequently observed on the surface of PP in marine debris [58].

Waves in the sea continuously apply stresses on plastic litters, leading to environmental stress cracking (ESC) [58]. PP degradation is initiated by autoxidation, leading to the onset of ESC. The initiation rate of autoxidation is relatively slow under sunlight irradiation. Delamination is likely to occur in the sea due to ESC [74]. The morphology of land-degraded PP (Land-PP) surfaces differs significantly from that of marine-degraded PP (Marine-PP) [58]. The conchoidal fractures shown in Table 3 can be explained by assuming they were generated through the ESC effect of waves. On the other hand, the surface texture of Land-PP consists of linear fracture, with no distinct step-like structures observed (see Table 2). This implies that Land-PP was exposed to ordinary weathering without ESC. Land-PP fragmentation is attributed to weathering and surface abrasion caused by wind and other factors. These results suggest that the fragmentation mechanisms differ between marine and terrestrial regions. The delamination mechanism in marine environment is based on ESC. Of course, this hypothesis requires thorough verification using models that mimic the actual phenomena. Autoxidation gradually progresses, promoting crack growth on the surface of PP sample. More precisely, crack formation is associated with the chemi-crystallization process, and autoxidation drives this process. Linear cracks, known as “linear fractures,” are frequently observed on the surfaces of both Land- and Marine-PPs and are a typical texture seen on the surface of PP debris.

To clarify the link between autoxidation and ESC, we note that autoxidation-induced chain scission reduces the molecular weight and mechanical toughness of polyolefins. This degradation lowers the critical stress threshold required to initiate ESC, especially under marine conditions where mechanical stress from waves is continuously applied. Therefore, autoxidation not only contributes to chemical degradation but also facilitates mechanical failure through ESC.

The cracks grow along the polymer flow lines, i.e., the orientation of the polymer chains [55,62]. This orientation generates a residual stress point that initiates cracks. An oriented surface region produces linear cracks, such as perpendicular crack lines and linear fractures on the PP surfaces in Table 2 and Table 3. Moreover, the degree of molecular orientation depends on the depth direction of the sample. The surface has a higher orientation degree than the inner part. Therefore, the orientation becomes weaker as autoxidation advances to deeper into the sample. When ESC occurs on the Marine-PP surface, the number and shape of cracks change depending on the degree of degradation. As a result, it is likely to exhibit complex textures such as conchoidal fractures. Of course, linear and conchoidal fractures also occur on PE surfaces through similar mechanisms. However, due to its smaller spherulite sizes and higher content of low-crystalline regions such as in LDPE, PE, particularly low-density grades with smaller spherulites, is less likely to exhibit these distinct textures as clearly as PP. Prolonged degradation of PP results in a mixture of trapezoidal and rectangular crack shapes, as shown in Table 3. These crack shapes represent crack propagation around PP spherulites. In our previous work, we have confirmed that the PP spherulite structure contributes to the formation of a trapezoidal, tile-like crack pattern [75]. In PP with a spherulitic base structure, the occurrence of trapezoidal crack patterns is inherent to its morphology. When the molecular orientation was relaxed by thermal pretreatment, only trapezoidal (tile-like) crack patterns were observed. Linear fractures occur due to orientation and are primarily observed in the early stages of degradation. However, as degradation progresses, trapezoidal crack patterns begin to appear together with linear ones. The unique surface texture of PP makes it easy to separate from other plastic waste. Developing sorting methods based on surface texture differences is expected to significantly reduce the cost of plastic recycling.

### 2.4. Low-Cost Waste Plastic Sorting Method Based on Crack Texture Differences

The future of PP recycling and marine plastic waste management lies in integrating advanced material characterization with artificial intelligence (AI)-driven sorting technologies [76,77]. One promising approach is the utilization of PP-specific crack patterns, which exhibit distinctive rectangular and trapezoidal geometries compared to other polyolefins, such as PE, and other types of plastics. These morphological signatures, formed during environmental degradation, can serve as robust visual markers for automated identification. AI-based image recognition technology, particularly convolutional neural networks (ConvNets), offers a highly scalable and cost-effective solution for real-time sorting of marine plastic debris. The distinctive surface texture of degraded PP appears to be sufficiently distinguishable using AI-based image recognition technology. Future research should prioritize creating comprehensive image datasets covering various degradation states, surface contamination levels, and lighting conditions. These datasets will enable the development of robust ConvNet models capable of achieving high classification accuracy under practical operational constraints.

In addition to their utility for sorting, surface cracks can also serve as qualitative indicators of aging in plastics exposed to environmental conditions. Crack morphology reflects the combined effects of photodegradation, mechanical stress, and chemical changes, making it a practical marker for assessing the degree of weathering.

Integrating this sorting approach with the mechanical recycling o PP can significantly enhance the circular economy by improving feedstock purity and reducing downstream processing costs. When coupled with life cycle assessment (LCA) analyses, these innovations can quantify environmental benefits, including substantial reductions in greenhouse gas emissions compared to the production of virgin PP. Additionally, deploying AI-enabled sorting modules in coastal waste management systems provides an opportunity for cost-effective, decentralized interventions that align with global sustainability goals. In summary, future work should focus on the following: (i) establishing standardized imaging protocols and annotated datasets; (ii) developing lightweight ConvNet architectures for on-site deployment; (iii) conducting LCA-based evaluations to demonstrate environmental and economic advantages; (iv) exploring hybrid strategies that combine AI-driven sorting with chemical and mechanical recycling pathways. These efforts will accelerate the transition toward a sustainable, circular plastics economy.

Although quantitative accuracy data for crack-based polymer identification is limited, PP and PE exhibit distinct surface morphologies under environmental degradation. As summarized in Table 2 and Table 3, PP typically shows linear fractures combined with rectangular or trapezoidal tiling, whereas PE tends to exhibit fragmentation, flakes, and cracks parallel to flow lines. These differences provide strong visual cues for polymer differentiation and can be leveraged for automated classification using AI-based image recognition. Future work should include systematic validation to establish accuracy metrics under real-world conditions.

## 3. Global Research Trends and Challenges

### 3.1. Rapid Expansion of Nanoplastic Research

Over the past decade, the field of MPs has expanded toward the nanoscale, yet fundamental issues persist regarding operational definitions, measurement harmonization, and realistic reference materials. Gigault et al. highlighted the semantic and metrological gap in defining “nanoplastics (NP)”, underscoring the need for size-resolved protocols and artifact control during sample preparation [20]. These factors are crucial for accurately and reproducibly handling NP samples. In parallel, wastewater and drinking-water studies emphasize occurrence and transformation processes, but suffer from limited comparability due to heterogeneous sampling, digestion, and detection workflows [74]. These constraints are especially apparent when applying insights from artificial systems to the open ocean environment. This is because salinity, solar radiation intensity, and biological activity in the open ocean combine to significantly alter degradation conditions. Further accumulation of data on meticulous degradation conditions is necessary.

### 3.2. Fragmentation Pathways: Terrestrial vs. Marine Contexts

A consistent theme emerging from field and laboratory observations is the difference in degradation rates between terrestrial and marine environments. Furthermore, studies of marine debris have reported characteristic morphological features, such as linear fractures, indentations and shell-like fractures, which reflect the combined effects of photochemical and mechanical stresses [56,72]. In coastal and offshore waters, chloride-mediated radical quenching retards autoxidation, altering crack initiation and growth kinetics and favoring texture preservation [32,58]. Conversely, in dry, oxygen-rich environments, terrestrial weathering promotes the accumulation of carbonyl groups, leading to brittle fracture [41,55]. Degradation may progress rapidly and manifest as transverse cracks perpendicular to flow lines. This is likely related to the presence of spherulites. Differences in degradation rates directly affect algorithmic sorting strategies utilizing realistic reference particle generation and texture characteristics. Due to the rapid progression of degradation, the distinct texture differences between degraded PP and PE on land may become blurred.

### 3.3. Outstanding Gaps and Standardization Needs

Three gaps impede progress: (i) calibrated reference MPs/NPs that represent polyolefin compositions and surface states formed under marine pH and salinity regimes [60]; (ii) depth-resolved degradation profiles linking orientation, spherulite size, and local radical flux to observed texture [43,61]; (iii) interoperable data standards for imaging and spectroscopy to enable robust model training and cross-study comparison [73,76]. Addressing these gaps will strengthen mechanistic inference and increase the reliability of AI-based identification approaches across field conditions. It is crucial to collect numerous discarded plastics through fieldwork and accumulate their surface image diagnostic data.

## 4. Analytical Techniques for Crack-Texture-Based Identification

### 4.1. Vibrational Spectroscopy (FT-IR/Raman)

FT-IR and Raman spectroscopy are essential for identifying polymers and tracking oxidation via carbonyl index (CI), vinyl groups, and CH stretching features. For PP and PE, CI progression correlates with photodegradation under controlled exposures. However, salinity and pH can alter radical pathways and thus decouple CI from crack propagation rates [40,41].

Surface-sensitive modes, such as FT-IR microscopy and Raman mapping, offer enhanced spatial resolution and are particularly useful for capturing chemical heterogeneity at crack initiation sites. Raman mapping, for example, visualizes oxidation heterogeneity at the micrometer scale and helps link chemical gradients to mechanical failure loci [49]. Despite their analytical power, these techniques are less frequently reported in the literature due to their cost and complexity.

While FT-IR and Raman spectroscopy are powerful tools for chemical typing, they do not directly quantify texture metrics such as crack spacing, aspect ratio, or trapezoid tiling density. Therefore, the most advanced polymer detection technologies often rely on material identification using infrared, laser, and X-ray fluorescence methods [78]. Recent developments also incorporate image recognition, sensor fusion, and target recognition via marking systems to improve sorting quality. Although FT-IR and Raman methods are central to these technologies, integrating them into waste plastic sorting systems significantly increases costs. Given the need to reduce sorting system costs, there is growing anticipation for the development of low-cost detection technologies, such as image diagnostics.

### 4.2. Electron Microscopy and Micro-Analytical Imaging

SEM coupled with EDS enables direct visualization of microcracks, delamination fronts, and spherulite-related boundaries while simultaneously detecting catalyst or filler residues that act as radical sinks or photocatalytic ROOH decomposition sites [27,54]. For blown-extruded PP, SEM reveals the canonical network of transverse cracks intersected by longitudinal linkers. Under seawater retardation, clear trapezoidal/rectangular tiling consistent with preserved orientation fields is observed [62,64]. These features provide high-signal morphology inputs for computer vision.

### 4.3. Hyperspectral and AI-Based Image Recognition

Recent work integrates hyperspectral cues with ConvNet-based classifiers and property prediction frameworks to increase sorting accuracy under contamination and variable lighting [76,77]. For marine PP, texture descriptors (e.g., crack angle distribution, tiling compactness, and flow-parallel line density) can be coupled with spectral fingerprints to resolve PP vs. PE vs. PS in mixed debris. Crucially, dataset curation must span degradation states (early linear fracture dominance to mixed trapezoid tiling), salinity histories, and biofilm coverages [13,14]. To improve classification accuracy, we recommend using a standardized imaging protocol that includes mutual calibration with FTIR/Raman labels, even though this can increase costs, to avoid domain drift during model deployment.

## 5. Environmental Impact and Risk Assessment

### 5.1. Ecological Interactions and Biofilm Dynamics

Biofilm formation on MPs modifies buoyancy, aggregation, and transport, and can catalyze particle–particle sticking with biogenic materials, reshaping residence times and exposure profiles [13,14]. For floating PP/PE, biofilms may also mask surface texture, complicating visual identification; however, mechanical stress from waves continues to drive ESC beneath biofilm layers, preserving diagnostic tiling patterns over longer intervals [56,58]. It will also be necessary to develop biofilm removal technology.

### 5.2. Trophic Transfer and Human Health Considerations

Evidence of microplastic transfer across trophic levels and potential human exposure via seafood consumption or atmospheric fallout underscores the need for size-resolved risk frameworks [12,21]. While toxicity mechanisms remain context-dependent—adsorbed pollutants, additives, and particle size/shape effects—fragmentation into smaller classes (including nanoplastics) likely increases bio-availability and cellular interaction potential [11,65]. Establishing reference PP/PE particles produced under marine-realistic conditions (pH, salinity, sunlight) will support reproducible toxicology and fate studies [19,60].

## 6. Future Perspectives

### 6.1. Roadmap for AI-Enabled Sorting

Short-term priorities include (i) open, annotated image repositories spanning texture states and biofilm levels; (ii) lightweight ConvNet architectures optimized for edge devices at coastal facilities; and (iii) cross-modal learning to fuse texture descriptors with spectral labels for robust classification under field variability. Crack pattern recognition models (such as convolutional neural networks, or ConvNets) tend to be less accurate when the conditions under which images are trained (e.g., lighting, background, biofilm adhesion, and regional variations) differ from actual field conditions. Domain adaptation (DA) is a technique designed to correct this “distribution discrepancy” or “domain shift.”

Medium-term efforts should develop DA pipelines to cope with seasonality, geography, and illumination shifts, and validate against reference particles generated with seawater-realistic protocols.

### 6.2. Policy, Standards, and Data Interoperability

Convergence toward interoperable standards for sampling, digestion, and detection will reduce uncertainty across studies and accelerate technology transfer to municipal and coastal waste systems. Harmonized reporting for texture metrics—crack spacing, orientation anisotropy, and trapezoid tiling density—alongside CI and vinyl indices will facilitate meta-analyses and improve risk modeling. There is a need to improve the international interoperability of data concerning the progression of these degradations.

### 6.3. LCA and Circular-Economy Integration

Integrating AI-enabled sorting with mechanical recycling of PP can increase feedstock purity and yield tangible climate benefits relative to virgin production; LCA studies should quantify emission reductions and resource savings under realistic contamination and biofilm scenarios. Hybrid strategies that combine biological, chemical, and mechanical routes for difficult streams (e.g., heavily weathered composites) warrant comparative evaluation, guided by texture-derived separability and degradation stage mapping.

### 6.4. Ecological Implications of Increased Surface Area

Increased surface roughness and crack formation during plastic aging significantly enhance the available surface area, which can facilitate biofilm development and potentially promote biodegradation. Moreover, these morphological changes increase the adsorption and absorption capacity for environmental contaminants, including hydrophobic organic compounds and heavy metals. While this review focuses primarily on crack patterns as indicators of degradation and sorting, the ecological implications of increased surface area represent an important aspect for future research.

## 7. Conclusions

This review emphasizes the important role of degradation mechanisms in MPs derived from polyolefins, such as PP and PE. Although autoxidation is the primary pathway for weathering degradation, environmental factors, particularly the presence of seawater, significantly influence its kinetics and by-products. Salinity and chloride ions inhibit radical formation, slowing photodegradation and altering crack propagation patterns. These effects result in distinct surface textures, such as rectangular or trapezoidal crack patterns in PP, which can serve as valuable indicators for identifying plastic types in marine debris.

These findings underscore the necessity of reevaluating conventional degradation models developed under terrestrial conditions because such models fail to account for the complex interplay of sunlight, seawater chemistry, and mechanical forces in marine environments. Understanding these interactions is essential for predicting MP formation rates, assessing environmental risks, and designing effective mitigation strategies. Additionally, the characteristic crack patterns observed on weathered plastics present promising opportunities for developing cost-effective sorting and recycling technologies, thereby contributing to improved resource recovery and reduced marine plastic pollution.

AI-based image recognition using ConvNets offers a scalable, low-cost solution for the real-time sorting of marine plastic debris. Future research should focus on creating comprehensive image datasets that cover various degradation states, contamination levels, and lighting conditions in order to improve the robustness of the model. Integrating this approach with PP mechanical recycling can enhance feedstock purity, reduce processing costs, and support circular economy goals. LCA can quantify environmental benefits, such as reduced greenhouse gas emissions compared to virgin PP production. Deploying AI-enabled sorting modules in coastal waste systems allows for decentralized, cost-effective interventions that align with global sustainability targets. Key priorities include:

(1) Standardized imaging protocols and annotated datasets;

(2) Lightweight ConvNet architectures for on-site use;

(3) LCA-based evaluations of environmental and economic impacts;

(4) Hybrid strategies combining AI sorting with chemical and mechanical recycling.

Finally, addressing outstanding gaps—such as harmonized definitions of NPs, depth-resolved degradation profiling, and global data interoperability—will accelerate the transition from descriptive studies to predictive, technology-enabled solutions. These efforts will not only improve resource recovery and reduce marine plastic pollution but also align with international sustainability and policy frameworks.

This review bridges fundamental degradation mechanisms with practical identification and recycling strategies. It provides a roadmap to advance scientific research, guide policy on marine plastic mitigation, and accelerate industrial adoption of AI-enabled sorting technologies for a circular economy.

## Figures and Tables

**Figure 1 molecules-30-04461-f001:**
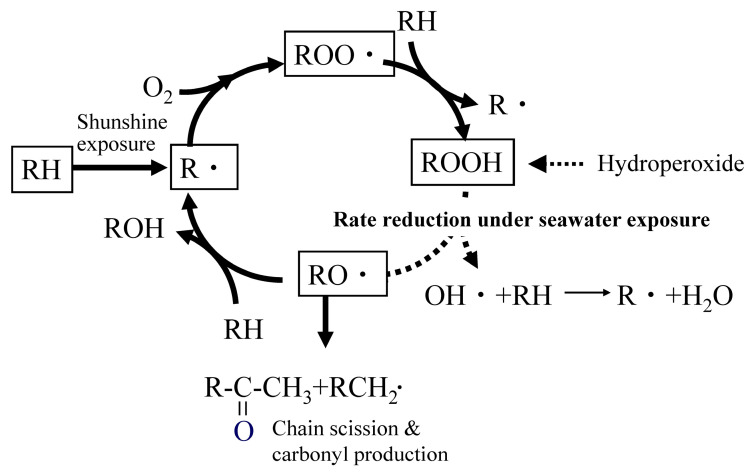
Autoxidation under seawater exposure.

**Figure 2 molecules-30-04461-f002:**
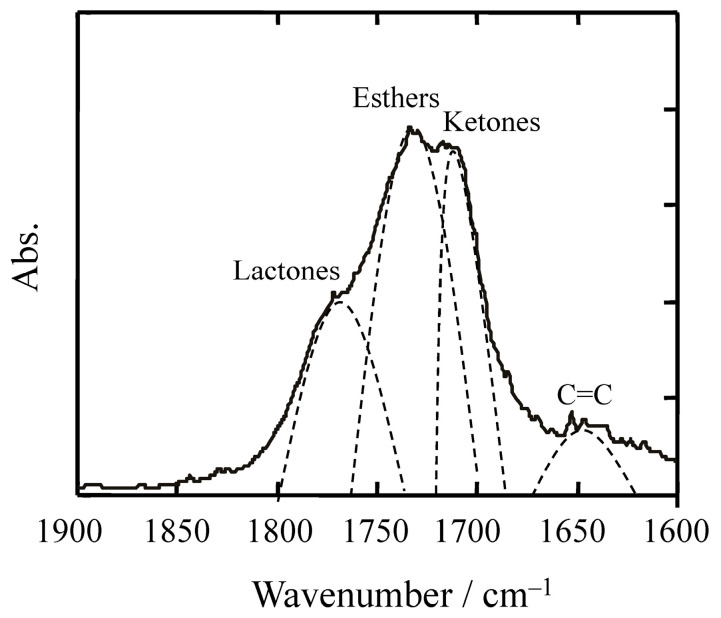
Example of an infrared spectrum of typical degraded polypropylene.

**Figure 3 molecules-30-04461-f003:**
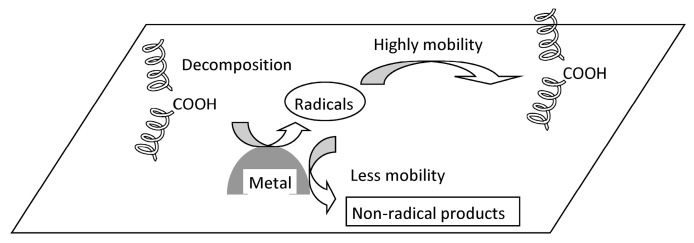
Local autoxidation mechanism in polyolefin.

**Figure 4 molecules-30-04461-f004:**
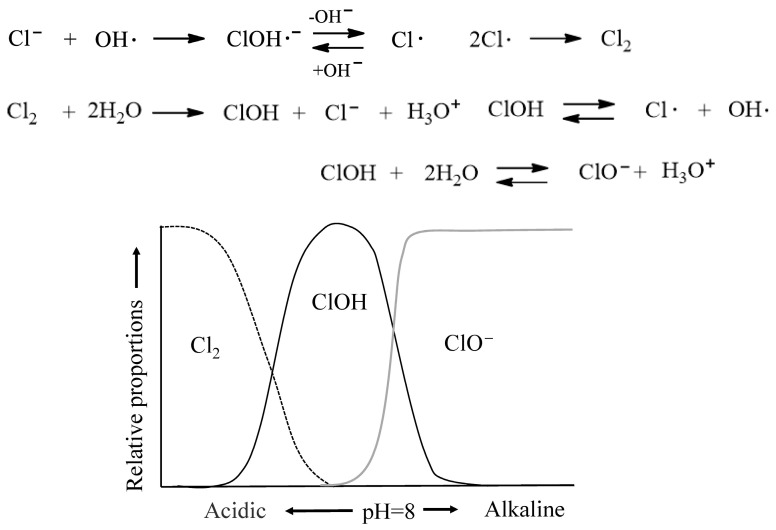
Mechanisms for transfer reaction to Cl^−^ in seawater and formation of chlorine-containing species in Cl_2_ solution as a function of pH.

**Table 1 molecules-30-04461-t001:** Key functional groups and reaction types observed during polyolefin decomposition.

Functional Group	Degradation Method	Types of Polymers	References
Esters	Photo, thermal Photo	PE PP	[39] [31,40,41]
Vinyl alkenes	Photo, thermal Photo	PE PP	[39] [31,40,41]
Acids	Photo, thermal Photo	PE PP	[39] [31,40,41]
Aldehydes	Photo, thermal	PE	[39]
γ-lactones	Photo, thermal Photo	PE PP	[39] [31,40,41]
Alcohol	Photo, thermal Photo	PE PP	[39] [40]
Other (crosslinking)	γ-radiation Photo Thermal	EP copolymer EP copolymer EP copolymer	[42] [43] [44]

**Table 2 molecules-30-04461-t002:** Crack texture on various weathered or photo-degraded plastic surfaces under different environmental conditions.

Enviro. Condition	Types of Polymer	Surface Texture	References
Photodegradation in air	PP	Network of the transverse cracks & longitudinal cracks linking the transverse cracks	[61]
Photodegradation in air	PP	Cracks perpendicular to flow lines	[57,62,63]
Natural weathering	PP	Cracks and grooves	[31]
Outdoor and accelerated laboratory weathering	PP	Cracks perpendicular to injection direction	[64]
Accelerated laboratory weathering in pure water	PP	Cracks perpendicular to extrusion direction & originating from spherulites (by blown-extrusion)	[24]
Accelerated laboratory weathering in pure water	PE(LDPE)	Cracks parallel to injection direction and/or fragmentation	[65]
Accelerated laboratory weathering in pure water	PE(LDPE)	Cracks perpendicular to extrusion direction (by blown-extrusion)	[24]
Accelerated laboratory weathering in air	Recycled PP and PE (LLDPE & HDPE)	Cracks propagating from edge toward center	[66]

**Table 3 molecules-30-04461-t003:** Crack texture on various weathered or photo-degraded plastic surfaces under seawater conditions.

Seawater Condition	Types of Polymer	Surface Texture	References
Beach	PP, PE	Conchoidal fractures	[56]
Beach	PE	Linear fracture, pit, groove	[56]
Beach	PP	Defined fracture	[72]
Photodegradation	PE	Flakes	[19]
Photodegradation	PP	Cracks and flakes	[19]
Accelerated laboratory weathering in seawater	PP	Trapezoidal and rectangular shapes	[73]

Note: Descriptions such as “flakes” are quoted directly from the original references to preserve the authors’ terminology.

## Data Availability

No new data were created or analyzed in this study. Data sharing is not applicable to this article.

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
