# Peer review of "Microplastic Formation and Surface Crack Patterns: A Method for Waste Plastic Identification"

_molecules, 2025, doi:10.3390/molecules30224461_

Round 1

Reviewer 1 Report

Comments and Suggestions for Authors

The manuscript “Microplastic Formation and Surface Crack Patterns: A Method for Waste Plastic Identification” provides a comprehensive review of the mechanisms leading to the formation of cracks and how they could be exploited. Overall, the review is useful and well written. Some details could be clarified or improved before publication.

Abstract: Provide the objective and details on the review approach (e.g., number of studies).

1. Introduction:

It would be interesting if the authors could provide details on the half-life of microplastics formation and removal during the review.

The review strategy was not mentioned in the introduction.

2. Degradation Reactions and Progression Behavior

Please address the following aspects in more detail:

-Rate of group formation. Formation of groups that can suffer further reactions and/or removal of groups that can be used by organisms.

-Identification of “markers” of degradation in polymers - including the carbonyl index (mentioned) but also problems arising from it, such as the confounding effect of  alkanes and alkenes from PE and fats in Pyr-GC-MS.

Could delamination be related to additives in plastic? Please address the influence of additives in plastic degradation.

PE samples from the environment are often “sticky” and break easily, looking melted, can you also address what causes these mechanical properties?

“Consequently seawater acts as a retarder of

autoxidation, thereby reducing the rate of photodegradation, or weathering degradation“ I remembered this study on the degradation of PE in seawater which might be relevant:

Degradation of polyethylene microplastics in seawater: Insights into the environmental degradation of polymers

(https://doi.org/10.1080/10934529.2018.1455381).

Can we also use surface cracks to define the aging of plastics in the environment?

The use of cracks to identify polymers in recycling seems interesting. Can you provide more details on the accuracy of the process? Also, can you provide a table with a summary on how to differentiated PE and PP based on surface properties?

Please introduce more paragraphs in text to improve readability. Also, consider adding images to illustrate the changed mentioned in plastics and/or changes to IR or Raman spectra.

3. Global Research Trends and Challenges

“Conversely, in dry, oxygen-rich environments, terrestrial weathering promotes the accumulation of carbonyl groups, leading to brittle fracture [43, 57]” Could you say that more microplastics are formed in terrestrial environments?

4. Analytical Techniques for Crack-Texture-Based Identification

Provide more details on what can be used as a measure of weathering in these techniques.

Can cracks also compromise the collection of spectra? For instance, areas near cracks may present different composition from the rest of the surface.

Can you suggest a reference plastics to produce in the laboratory? For instance, to acquire pellets and expose them for X days to the sun or to UV lamps.

5. Environmental Impact and Risk Assessment

I think one aspect missing is the increase in surface area with the cracks, which facilitate biofilm formation (or even biodegradation) and adsorption/absorption of environmental contaminants to the plastic surface.

6. Future Perspectives

This seem to be repeated with previous sections and conclusions. Maybe this section could be suppressed and text joined with the rest, to avoid repetitions.

Maybe include some information on how to study the degradation of polymers and a table summarizing current results on the half-life.

Author Response

Reviewer 1

Comment 1. Provide the objective and details on the review approach (e.g., number of studies).

Answer: The text was added and revised from lines 88 to 93 of the introduction.

Comment 2. It would be interesting if the authors could provide details on the half-life of microplastics formation and removal during the review. The review strategy was not mentioned in the introduction.

Answer: The text was added and revised from lines 93 to 98 of the introduction.

Comment 3. 2. Degradation Reactions and Progression Behavior

Comment: Please address the following aspects in more detail: Rate of group formation. Formation of groups that can suffer further reactions and/or removal of groups that can be used by organisms.

Answer: The following text has been added from lines 150 to 156: Although the formation of functional groups such as esters, acids, and vinyl alkenes during polyolefin degradation has been documented (see Table 1), the quantitative rate of their formation and the subsequent reactions—such as removal or utilization by organisms—remain largely unknown. Current studies provide qualitative evidence of side reactions but do not offer kinetic data or ecological assimilation pathways. Therefore, a detailed discussion on these aspects is beyond the scope of this review and represents an important area for future research.

Comment : Identification of “markers” of degradation in polymers - including the carbonyl index (mentioned) but also problems arising from it, such as the confounding effect of alkanes and alkenes from PE and fats in Pyr-GC-MS.

Answer: The text has been added from lines 156 to 164. Additionally, new Figure 2 has been added to provide further clarification.

Comment: Could delamination be related to additives in plastic? Please address the influence of additives in plastic degradation.

Answer: Delamination in degraded plastics is indeed influenced by the presence and transformation of additives. In particular, degradation does not typically initiate until stabilizing additives—such as antioxidants—are either consumed or chemically altered. Therefore, delamination is unlikely to occur unless these additives lose their protective function. However, in the case of post-consumer plastic waste, the type and residual amount of additives vary significantly depending on the original application and history of the material. This variability makes it extremely difficult to generalize the influence of additives on degradation behavior. For this reason, the present review does not attempt to systematically address the role of additives in plastic degradation.

Comment: PE samples from the environment are often “sticky” and break easily, looking melted, can you also address what causes these mechanical properties? Consequently seawater acts as a retarder of autoxidation, thereby reducing the rate of photodegradation, or weathering degradation“ I remembered this study on the degradation of PE in seawater which might be relevant:

Degradation of polyethylene microplastics in seawater: Insights into the environmental degradation of polymers

(https://doi.org/10.1080/10934529.2018.1455381).

Da Costa JP, Nunes AR, Santos PSM, Girão AV, Duarte AC, Rocha-Santos T. Degradation of polyethylene microplastics in seawater: Insights into the environmental degradation of polymers. J. Environ. Sci. Health. A Tox. Hazard. Subst. Environ. Eng. 2018 Jul 29;53(9):866-875 https://doi.org/10.1080/10934529.2018.1455381 

Answer: The corresponding text was added and revised from lines 257 to 259, and cited new Ref. 73.

Comment: Can we also use surface cracks to define the aging of plastics in the environment?

Answer: Yes, surface cracks can serve as indicators of aging in plastics exposed to environmental conditions. Crack morphology reflects the combined effects of photodegradation, mechanical stress, and chemical changes, making it a useful qualitative marker for assessing the degree of weathering. The corresponding text was added and revised from lines 340 to 343.

Comment: The use of cracks to identify polymers in recycling seems interesting. Can you provide more details on the accuracy of the process? Also, can you provide a table with a summary on how to differentiated PE and PP based on surface properties?

Answer:  We believe that the table summarizing methods for distinguishing PE and PP based on surface characteristics has already been presented, as it is essentially a compilation of Tables 2 and 3. Since it contains redundant content, adding it as a new table would make this review redundant. Instead, we added the following text to lines 357–364: Although quantitative accuracy data for crack-based polymer identification are limited, PP and PE exhibit distinct surface morphologies under environmental degradation. As summarized in Tables 2 and 3, PP typically shows linear fractures combined with rectangular or trapezoidal tiling, whereas PE tends to exhibit fragmentation, flakes, and cracks parallel to flow lines. These differences provide strong visual cues for polymer differentiation and can be leveraged for automated classification using AI-based image recognition. Future work should include systematic validation to establish accuracy metrics under real-world conditions.

Comment: Please introduce more paragraphs in text to improve readability. Also, consider adding images to illustrate the changed mentioned in plastics and/or changes to IR or Raman spectra.

Answer: Following your suggestion, we added several paragraphs. Additionally, we included the IR spectrum of the degraded PP as a new Figure 2. Raman spectroscopy is significantly less sensitive to polar functional groups than IR spectroscopy and shows fewer changes during degradation. Thus, the IR spectrum provides a clearer model and is easier to interpret. For this reason, we have chosen to present only the IR spectrum.

Comment 4. 3. Global Research Trends and Challenges

“Conversely, in dry, oxygen-rich environments, terrestrial weathering promotes the accumulation of carbonyl groups, leading to brittle fracture [43, 57]” Could you say that more microplastics are formed in terrestrial environments?

Answer: Yes, we believe that the conditions for generating microplastics are better on land than in the ocean. Consequently, larger quantities of microplastics are being produced.

Comment 5. 4. Analytical Techniques for Crack-Texture-Based Identification

Provide more details on what can be used as a measure of weathering in these techniques.

Answer: While techniques such as FT-IR and Raman spectroscopy can track chemical changes (e.g., carbonyl index, vinyl group formation) and visualize oxidation heterogeneity, quantitative measures of weathering beyond these indicators are not yet standardized. Current studies provide qualitative evidence rather than robust metrics, and developing reliable measures remains an open research challenge. As this review aims to summarize existing knowledge, further details cannot be provided beyond what is currently reported in the literature.

Comment: Can cracks also compromise the collection of spectra? For instance, areas near cracks may present different composition from the rest of the surface.

Answer: The edges of cracks are likely to be more susceptible to oxidation because two surfaces are exposed to air. However, for IR spectroscopy, which is highly sensitive to polar functional groups such as carbonyls, the beam spot size required for reliable measurement is typically ≥10 μm (commonly around 100 μm). Therefore, accurately detecting differences in peak intensity of polar functional groups between crack edges and other surface regions is extremely challenging.

Comment: Can you suggest a reference plastics to produce in the laboratory? For instance, to acquire pellets and expose them for X days to the sun or to UV lamps.

Answer: If you desire accelerated degradation, our recently developed method [Nakatani et al., Sci Rep 13, 4247 (2023). https://doi.org/10.1038/s41598-023-31488-w] would be of use. For PP, you can obtain sufficiently degraded samples in about 3 to 6 days.

Comment 6. 5. Environmental Impact and Risk Assessment

I think one aspect missing is the increase in surface area with the cracks, which facilitate biofilm formation (or even biodegradation) and adsorption/absorption of environmental contaminants to the plastic surface.

Answer: Following your suggestion, we added a new section as” 6.4. Ecological Implications of Increased Surface Area .” Please see the section!

Comment 7. 6. Future Perspectives

This seem to be repeated with previous sections and conclusions. Maybe this section could be suppressed and text joined with the rest, to avoid repetitions.

Answer: We appreciate the suggestion. However, Section 6 is intended to provide a consolidated view of future perspectives, highlighting gaps in current knowledge and technologies that need to be addressed. While some overlap with previous sections and conclusions is inevitable, removing this section or dispersing its content would make it more difficult for readers to grasp the overall roadmap for future research and technological development. For clarity and completeness, we believe maintaining this section adds value to the review.

Comment 8: Maybe include some information on how to study the degradation of polymers and a table summarizing current results on the half-life.

Answer: We appreciate the suggestion. However, the main focus of this review is on crack patterns as a method for identifying waste plastics, rather than on the kinetics of polymer degradation. While degradation is discussed as a key factor influencing crack formation, providing detailed methodologies for studying degradation or summarizing half-life data would shift the scope of this review away from its intended purpose. For clarity and relevance, we have limited our discussion to mechanisms and analytical techniques directly related to crack development and identification.

Reviewer 2 Report

Comments and Suggestions for Authors

This review synthesizes research on MP formation from PP/PE, contrasting degradation in marine and terrestrial environments. The central contribution is identifying that salinity and chloride ions in seawater inhibit autoxidation, the primary degradation pathway. This retardation of chemical degradation allows mechanical and morphological factors to dominate crack propagation. This results in distinctive surface patterns which are less prevalent in land-weathered plastics. The authors propose these unique "crack textures" can be used as a reliable, low-cost identification method for plastic sorting, particularly using AI-based image recognition.

  1. The phrase "molding-induced orientation and superstructures" is slightly general. For greater clarity, consider specifying "...such as polymer chain orientation and spherulite structures," which aligns with the details provided later in the text and immediately grounds the reader.
  2. The dotted arrow for "Rate reduction under seawater exposure" points from the ROOH decomposition step. However, the text strongly suggests the primary inhibition mechanism is the quenching of OH· radicals by Cl-. Please ensure the diagrammatic representation in Figure 1 accurately and clearly reflects the primary mechanism described in the text.
  3. The title of Table 1, "Compounds produced during polyolefin decomposition," could be more precise. The entries listed are primarily functional groups or classes of compounds, and 'Crosslinking' is a reaction type. Consider retitling to "Key functional groups and reaction types observed during polyolefin decomposition" for improved terminological accuracy.
  4. The link between autoxidation and ESC is key. The argument would be strengthened by explicitly stating that autoxidation-induced chain scission lowers the polymer's molecular weight and toughness, thereby reducing the critical stress required to initiate ESC in the presence of the marine environment.
  5. The phrase "some PE is less likely to exhibit these textures" is imprecise. Please clarify this by linking it more directly to the preceding cause. For example: "...consequently, PE, particularly low-density grades with smaller spherulites, is less likely to exhibit these distinct textures as clearly as PP."
  6. In Table 3, the descriptions "Flakes" and "Cracks and flakes" are less descriptive of a crack pattern than "Trapezoidal and rectangular shapes". To improve parallelism with Table 2, please clarify if 'flakes' refers to a specific crack morphology or if it is a different type of degradation product.
  7. When discussing FT-IR/Raman for tracking oxidation, it would be beneficial to briefly specify the use of surface-sensitive modes. This reinforces the link between the surface crack patterns and the surface chemical changes, as bulk-mode analysis may not capture the heterogeneity discussed in Section 2.2.

Author Response

Reviewer 2

English could be improved to more clearly express the research.

Answer: English has been reviewed by a native speaker, and any issues with expression have been thoroughly improved.

Comment 1: The phrase "molding-induced orientation and superstructures" is slightly general. For greater clarity, consider specifying "...such as polymer chain orientation and spherulite structures," which aligns with the details provided later in the text and immediately grounds the reader.

Answer: Based on your suggestion, we have made the following corrections:  polymer chain orientation and spherulite structures

Comment 2: The dotted arrow for "Rate reduction under seawater exposure" points from the ROOH decomposition step. However, the text strongly suggests the primary inhibition mechanism is the quenching of OH· radicals by Cl-. Please ensure the diagrammatic representation in Figure 1 accurately and clearly reflects the primary mechanism described in the text.

Answer: Thank you for your comment regarding Figure 1. We would like to clarify that Figure 1 is intended to depict the canonical autoxidation mechanism of polyolefins, which proceeds via ROOH decomposition and radical propagation. The inhibitory effect of Cl⁻ in seawater, as described in the text, operates upstream by quenching OH· radicals generated by sunlight, thereby suppressing the initiation of autoxidation. This environmental modulation is discussed in detail in the manuscript and illustrated separately in Figure 4. Therefore, we respectfully believe that Figure 1 does not require modification, as it accurately represents the intrinsic chemical mechanism of autoxidation.

Comment 3: The title of Table 1, "Compounds produced during polyolefin decomposition," could be more precise. The entries listed are primarily functional groups or classes of compounds, and 'Crosslinking' is a reaction type. Consider retitling to "Key functional groups and reaction types observed during polyolefin decomposition" for improved terminological accuracy.

Answer: As you pointed out, we revised the title of Table 1.

Comment 4: The link between autoxidation and ESC is key. The argument would be strengthened by explicitly stating that autoxidation-induced chain scission lowers the polymer's molecular weight and toughness, thereby reducing the critical stress required to initiate ESC in the presence of the marine environment.

Answer: The following text was added from lines 295 to 300: To clarify the link between autoxidation and environmental stress cracking (ESC), we note that autoxidation-induced chain scission reduces the molecular weight and mechanical toughness of polyolefins. This degradation lowers the critical stress threshold required to initiate ESC, especially under marine conditions where mechanical stress from waves is continuously applied. Therefore, autoxidation not only contributes to chemical degradation but also facilitates mechanical failure through ESC.

Comment 5: The phrase "some PE is less likely to exhibit these textures" is imprecise. Please clarify this by linking it more directly to the preceding cause. For example: "...consequently, PE, particularly low-density grades with smaller spherulites, is less likely to exhibit these distinct textures as clearly as PP."

Answer: As you pointed out, we have revised the following text from lines 312 to 313:  consequently, PE, particularly low-density grades with smaller spherulites, is less likely to exhibit these distinct textures as clearly as PP.

Comment 6: In Table 3, the descriptions "Flakes" and "Cracks and flakes" are less descriptive of a crack pattern than "Trapezoidal and rectangular shapes". To improve parallelism with Table 2, please clarify if 'flakes' refers to a specific crack morphology or if it is a different type of degradation product.

Answer: Thank you for your suggestion regarding the terminology used in Table 3. We would like to clarify that the descriptors "flakes" and "cracks and flakes" are directly taken from the original references cited in the table. To maintain fidelity to the source literature, we have retained the original wording. While these terms may appear less specific than "trapezoidal and rectangular shapes," they reflect the terminology used by the respective authors to describe the observed degradation features. We have added a brief note in the table caption to clarify this point.

Comment 7: When discussing FT-IR/Raman for tracking oxidation, it would be beneficial to briefly specify the use of surface-sensitive modes. This reinforces the link between the surface crack patterns and the surface chemical changes, as bulk-mode analysis may not capture the heterogeneity discussed in Section 2.2.

Answer: Thank you for your insightful comment. We agree that surface-sensitive modes, such as FT-IR microscopy and Raman mapping, are particularly valuable for capturing the chemical heterogeneity associated with surface crack patterns. While such techniques provide high spatial resolution, they are less commonly employed due to their cost and time requirements, and only a limited number of studies have utilized them. Nevertheless, we have added a brief clarification in the manuscript to acknowledge the importance of surface-sensitive approaches in linking chemical changes to morphological features. We have revised “4.1. Vibrational Spectroscopy (FT-IR/Raman) “ section.

Round 2

Reviewer 1 Report

Comments and Suggestions for Authors

The authors have addressed all comments. The manuscript can now be accepted. I believe this work is of importance to the field by addressing an understudied aspect of degradation.